# Literature Review of an Anterior Deprogrammer to Determine the Centric Relation and Presentation of Cases

**DOI:** 10.3390/bioengineering10121379

**Published:** 2023-11-30

**Authors:** Maria Danko, Lubos Chromy, Norbert Ferencik, Marcela Sestakova, Petra Kolembusova, Tomas Balint, Jaroslav Durica, Jozef Zivcak

**Affiliations:** 1Department of Biomedical Engineering and Measurement, Faculty of Mechanical Engineering, Technical University of Kosice, 04200 Kosice, Slovakia; lubos.chromy@tuke.sk (L.C.); norbert.ferencik@tuke.sk (N.F.); petra.kolembusova@tuke.sk (P.K.); tomas.balint@tuke.sk (T.B.); jozef.zivcak@tuke.sk (J.Z.); 21st Department of Stomatology, Faculty of Medicine, Pavol Jozef Safarik University in Kosice, 04154 Kosice, Slovakia; marcela.sestakova@gmail.com (M.S.); jaroslav.durica@upjs.sk (J.D.)

**Keywords:** anterior deprogrammer, bruxism, occlusal splint, night guard, 3D printing

## Abstract

The increasing demand for dental aesthetics, articulation corrections, and solutions for pain and frequent bruxism demands quick and effective restorative dental management. The biomedical research aimed to create a beneficial, ecological, and readily available anterior deprogrammer to determine the centric relation (CR) of cases. This medical device is additively manufactured from a biocompatible material. Size is customizable based on the width of the patient’s anterior central incisors. This is a pilot study with two subjects. The task was to develop a complete data protocol for the production process, computer-aided design (CAD), and three-dimensional (3D) printing of the anterior deprogrammers. The research focused on creating simple and practically applicable tools for the dentist’s prescription (anterior deprogrammer in three sizes), and secondly for the communication between the dentist and the patient (computer application). The tested hypothesis was whether, according to these novel tools, it is possible to produce functional occlusal splints, which could be manufactured using current technologies. This study compared a traditional splint with a digitally designed and 3D-printed one. The tested hypothesis was whether manufactured occlusal splints differ in patients’ subjective perception of comfort. Each conservative treatment was monitored for ten weeks. Initial results are promising; no statistically significant difference was found between the productive technologies.

## 1. Introduction

The rising trend for dental and maxillofacial aesthetics, even in the case of temporomandibular disease (TMD) [1,2], creates enormous pressure on dental clinics and dental technicians. The worldwide effort for an ecological approach seeks solutions by reducing the carbon footprint and production waste. In this context, mnemonic and digital visual aids are coming to the forefront, facilitating the understanding of the broader context of the human body and its interaction with the action of internal and external forces [2].

Based on these sought-after quantities, biomedical research was created regarding anterior deprogrammers for the upper and lower jaw designed to solve the unbalanced function of the temporomandibular joint (TMJ) [3,4] and masticatory muscles. Temporomandibular disorders (TMDs) involve several factors or causes [1], representing a heterogeneous group of conditions concerning the TMJ and the surrounding structures [2,5,6]. Symptoms include jaw clicking, muscle tenderness, pain during opening, and limited mouth opening [1,7]. This disease’s etiology is multifactorial and is made up of biological factors [8] (e.g., internal derangements in the TMJ), psychological factors (e.g., depression, anxiety, and stress), and social factors (e.g., a learned response to pain). Psychological influences are considered highly significant in the development of TMD [1,2]. The results of some studies’ qualitative analyses have shown a positive association between migraines and TMDs (general) and an absence of an association between articular TMD without pain [2,4,8] (disk displacement and degenerative diseases) and primary headaches. The meta-analysis results confirmed a positive association between both pain-related TMD and migraine and chronic headaches. The association between pain-related TMD and tension-type headaches is controversial [4].

Many current studies [5,6] consider TMD to be associated with an imbalance of the whole body [9,10], and others [3,7,8] deal with the impact of TMD on the quality of a our sleep. Current research focuses on a conservative solution through a night occlusal splint [8,9,11,12,13] and considers it a successful treatment, as well as physiotherapy. There are several ways to manage the closed-lock temporomandibular joint disorder conservatively. These include educating and counseling the patient, manipulating the mandible, using a splint, exercise therapy, and pharmacotherapy [14]. Conservative treatment should always be the first choice, be minimally invasive, and include motivation towards the use of occlusal splint therapy or physiotherapy [15,16]. Surgery should only be considered after conservative management fails [17,18]. However, in complex cases, combining occlusal splint and arthrocentesis is more effective for joint disc dislocation without reduction [19,20]. On the other hand, some studies have found no positive effects on condylar pathways [9,18,21] when additional arthrocentesis was applied alongside splint therapy in patients with internal derangement of the TMJ. Reviews often point out limitations in studies, which gives rise to the need for further research [10,22,23,24,25,26,27,28]. These limitations may include small sample sizes, a lack of randomization, inadequate blinding methodologies, poor concealment of allocation, improper handling of withdrawals and losses, selective or incomplete reporting, short test periods, the absence of a control group, inadequate baseline data, and shortcomings in outcome reporting [10,22,24,28]. The application of splints should therefore be considered at an individual patient’s level [29]. This may be due to the multifactorial nature of the disease. A thorough understanding of occlusion [30], phonetics, esthetics, and laboratory steps is also necessary [25,31]. Occlusal splints, known as night guards, orthotics devices, oral appliances, or modified nociceptive trigeminal inhibition splints [32], are used routinely in dental management [14,15]. Occlusal splints help reduce tension, decrease muscle activity [30], and prevent harmful effects caused by bruxism and TMDs [15,33]. They are a tool for treating patients with TMD and bruxism for occlusal stabilization [34] and for reducing dentition wear [15,35,36].

The biomechanical influence of the device: To define the geometric dimensions of the night guard, a unique tool is needed to determine the centric relation. However, the available medical aids do not always meet the treatment requirements [15,16]. The discussion centers on the requirements imposed upon front deprogrammers in relation to the dimensions associated with the width of the central incisors as well as the fixation in modeling material. It is essential to ensure that front deprogrammers adhere to these requirements to guarantee an accurate and effective outcome.

Based on the requirements of the dental practice, research that combines biomedical engineers and dental technics collaborating was created in collaboration with doctors from a dental clinic specializing in conservative dentistry and prosthetics. This joint activity focuses primarily on setting the centric relation (CR) of the upper jaw (maxilla) and the lower jaw (mandible). The so-called centric relation defines the balance. According to Dawson, regardless of the vertical dimension or tooth position, the CR is the connection between the mandible and maxilla when the condyle–disk assemblies are correctly aligned and in the most superior position against the eminent [14,32,37]. The condyle–disk assemblies are reinforced medially in their highest position. Centric relation is, therefore, also in the middle [14,30]. Condyle–disk assemblies that are centrally aligned can withstand the maximum loading applied by the elevator muscles without experiencing any pain [38].

The research effort is to find a simple and effective solution applicable in practice with the right balance between the function of the masticatory muscles, the anatomical relationship of the condyle–disc-articular socket in the temporomandibular joint, and the relationship between the teeth of the upper and lower dentition, regardless of the etiology of the disease [1].

Based on the above, the primary goal is to design and manufacture a modifiable anterior deprogrammer. Considering ecological principles, an additive production method based on CAD data of a 3D-designed model [39] was used. The tested hypothesis was whether, according to this medical device, it is possible to produce functional occlusal splints, which could be manufactured using any current technology. Another hypothesis that was tested was whether manufactured occlusal splints differ in patients’ subjective perception of effectiveness and comfort. For this purpose, it was necessary to create a new procedural algorithm considering the multifactorial nature of TMD to monitor the patient’s subjective perception of comfort. Based on previous research, it can be proven that temporomandibular joint pain goes beyond the physiology of articulation and is significantly influenced by the external and internal influences of the environment in the human body [9,10].

## 2. Patients, Materials, and Methods

### 2.1. Patients

Two women probands (*n* = 2) were included. The same signs were experienced by the patients: pain in the temporomandibular joint area, stiffness of the masticatory muscles, and limited range of motion in the temporomandibular joint [17]. Objective examination: symmetrical faces without skin disorders, deformities, swellings, and skin efflorescence. Via palpation, the sensitivity of the masticatory muscles was present in the area of mm. temporalis et masseter bilaterally. Opening–closing cycle with sound phenomena, without significant deviation of the lower jaw, without limit in function. Patients’ dentitions were treated. Dentitions did not show signs of significant abrasion [40]. Result from cone-beam computed tomography for the temporomandibular joint (CBCT TMJ: an asymmetric position of the articular heads was visible in coronal and axial sections (Figure 1 and Figure 2).

### 2.2. Materials

Production of an occlusal splint was possible using a digital workflow. Biocompatible Photopolymer Resin Dental LT Clear (Formlabs Inc., Somerville, MA, USA) was used to manufacture each of the anterior deprogrammers and for the occlusal splint for one volunteer (proband No. 2). A photocomposite resin is a type of dental resin used in restorative dentistry for tooth-colored restorations, such as dental fillings or bonding.

Photocomposite resins consist of a mixture of resin matrix and filler particles [41]. The resin matrix typically comprises a monomer or a combination of monomers, while the filler particles provide strength, wear resistance, and aesthetic properties. The filler particles are often made of glass, quartz, or ceramic materials. When the photocomposite resin is applied to the tooth cavity or surface, it can be shaped and molded to match the natural tooth anatomy. After achieving the desired shape, a dental curing light initiates a chemical reaction within the resin, causing it to harden and become solid. This process is known as polymerization. The material offers several advantages in restorative dentistry. These resins can be color-matched to the natural tooth shade, providing an aesthetically pleasing result. They bond well to the tooth structure, providing good adhesion and sealing properties.

Additionally, they can be easily shaped and polished to achieve a smooth and natural-looking restoration. It is worth noting that various types and brands of photocomposite resins are available, each with different properties and indications. The selection of a specific photocomposite resin depends on factors such as the clinical situation, the desired aesthetic outcome, and the practitioner’s preferences.

The selected Photopolymer Resin Dental LT Clear is ideal for rigid splints, occlusal guards, and other direct-printed long-term orthodontic appliances because it is a Class Iia long-term biocompatible resin. Post-curing required. It is a material suitable for use over an extended period on mucosal and skin surfaces. Dental materials must be biocompatible, especially when they will be in close contact with tissues for extended periods of time. Technical testing is utilized to evaluate a material’s biocompatibility and guarantee that it is safe for human usage [42,43]. Dental LT Clear is tested at NAMSA, Chasse sur Rhône, in France, and is certified biocompatible per EN-ISO 10993-1:2009/AC:2010. The product follows ISO standards [44,45,46]: EEN ISO 1641:2009; EN-ISO 10993-1:2009/AC:2010; EN-ISO 10993-3:2009; EN-ISO 10993-5:2009; EN 908:2008.

This resin has a high translucency and a nonlinear compression resistance of up to 600 N, equivalent to our biting force [41,44,47]. Teeth are the most resilient human tissues [48]. The 3D-printed splints, therefore, need post-curing [49]. Second, foods and drinks can impact the characteristics of intraoral appliances, reducing their aesthetic appeal and jeopardizing their structural integrity. Mouth conditions can also affect a material’s characteristics [50].

### 2.3. Methods

After establishing the diagnosis, the dentist used an anterior deprogrammer to create an interocclusal centric relation record [51]. Various clinical techniques can be employed to register the mandible in centric relation. The anterior stop is a helpful tool in identifying the centric relation position [52,53]. Following a confirmatory CT examination in this study, the dentist utilized a dimensionally adequate anterior deprogrammer to induce a physiologically correct relation between the upper and lower jaw. The participants were instructed to apply a moderate force on the deprogrammer device while rapidly protruding and retruding their mandible. Following this, they were required to maintain a centered relation position of their lower jaw for a period of time. Subsequently, occlusal registration material between the posterior teeth (only) was applied [51]. This article does not mention the traditional procedure to produce splints (Figure 3). It will focus more on a digital workflow [54] using artificial intelligence to facilitate the interpretation and reproduction of data.

Tools and equipment used in this study include SolidWorks 2022 (3D CAD design software, analysis software, and product data management software), 3D printer Formlabs Form 3B with technology LFS (low-force stereolithography), biocompatible material Dental LT V1 (Formlabs, USA), SW PreForm (Formlabs, USA), isopropyl alcohol (96%), UV lamp Form Cure (Formlabs, USA), pliers, micromotor Saeshin Forte 200α (Saeshin, Daegu, Republic of Korea), SW Zirkonzahn. Modellier, and dental scanner Zirkonzahn S900 ARTI Scanner (Zirkonzahn, Neuler, Germany).

Description of a CAD technique and 3D printing for fabricating an additively manufactured dental device using a complete digital workflow [54,55]: Stereolithography (SLA) 3D printing on the Formlabs Form 3B printer was chosen to manufacture all deprogrammers and one splint. Form 3B printer is specifically designed for dental and healthcare applications, which involves using a laser to selectively cure a liquid resin, creating solid layers that build up to form a 3D object. This technology allows for high-resolution prints with smooth surface finishes. The printer has a build volume of 145 × 145 × 185 mm. Form 3B offers high accuracy with a layer thickness ranging from 25 to 300 microns, producing detailed and precise parts. The main reason we used the Form 3B printer is because of its ability to produce a range of biocompatible dental materials formulated explicitly for dental applications, including surgical guides, splints, models, and more [43].

#### 2.3.1. Additive Manufacturing Protocol of the Anterior Deprogrammer

The 3D printing algorithm (Figure 4) is composed of the following protocol: (1) Pre-processing: data were processed for anterior deprogrammer to create CAD designs in SW SolidWorks in three sizes according to the width of central incisors; export to mesh (STL creation); import an STL file into SW PreForm for data optimization. (2) Processing: selection of a suitable material (biocompatible material Dental LT); model preparation for printing; precise positioning of 3D deprogrammer model for the CAD coordinate system; support structure design: full raft, density 0.50, touchpoint size 0.35 mm, layer thickness 0.100 mm; control of the main production parameters, i.e., the available material, choice of material, and wall thickness; setting of print parameters; data export to the 3D printer and the start of the production cycle; 3D printing. (3) Post-processing: immersion of printed models in isopropyl alcohol (96%) for 10 min to remove excess material; drying models with compressed air; insertion into a Form Cure UV lamp (Formlabs, Berlin, Germany), where they were additionally cured for 20 min at a wavelength of 405 nm and a temperature of 60 °C; mechanical removal of the supporting structure; additional processing of the surface of the models using a Saeshin Forte 200α handheld micromotor (Saeshin, South Korea); sandblasting of jig models for better fixation in silicone during impression making.

#### 2.3.2. Additive Manufacturing Protocol of the Occlusal Splint

Three-dimensional printing pre-processing. (1) Scan plaster models of teeth with the 3Shape E2 dental 3D scanner (3Shape, Denmark). Photograph upper and lower dentition models, and scan the jaw relationship using a bite made with a jig. (2) CAD design of bite splint in SW 3Shape. (3) Export to mesh (STL creation). (4) Import data into SW PreForm (Formlabs, USA). (5) Prepare the model for printing. Precisely position 3D jig model for the CAD coordinate system. (6) Support structure design: full raft, density 0.50, touchpoint size 0.35 mm, layer thickness 0.100 mm. (7) Control of the main production parameters, i.e., the available material, choice of material, and wall thickness. (8) Export data to the 3D printer and the start of the production cycle (Figure 5).

#### 2.3.3. Procedural Algorithm of Occlusal Splint Testing

A simple computer application (Figure 6a) was created to simplify patient, doctor, and operator communication [56,57]. The application was created in the Visual Studio environment in the C# programming language. It is an application based on Windows Form Application with a user interface. It contains an evaluation part, a communication part, and a backup part and uses server storage to store communication data between the patient, the doctor, and the operator. A firewall and general protection secure the data, as the server is located on the university network and is securely stored on physical storage without the possibility of cloud backup. The application has versatile use in testing medical devices and is the property of the Department of Biomedical Engineering and Measurement. 

The night occlusal splints were tested by patients in vivo, daily, for ten weeks. The meaning and purpose of the ten-point comfort rating scale were explained to the probands. The lower night limit of conservative treatment was set at six hours. If this limit was not observed, the patient had to rate the splint as intolerable. The comfort rating scale was set from 0 to 10 (0–1, non-limiting; 2–3, a slightly unpleasant feeling; 4–6, uncomfortable; 7–9, severe discomfort; 10, intolerable; Figure 6b).

## 3. Results

The sample size was calculated based on the statistical requirements for a population size of 54 patients (sample size: *n* = 18, confidence level: 90%, margin of error: ±10%, population proportion: 10% [58]). This research was a pilot case study involving two probands who tested the level of the procedural algorithm and were willing to send feedback daily.

We designed the anterior deprogrammer in three sizes (x = 12.6, y = 18.5, z = 9.9), (x = 14.0, y = 18.5, z = 9.9), (x = 16.0, y = 18.5, z = 9.9) [mm]. For this study, we used a medium-sized device.

All calculations were performed using statistical data analysis. The Shapiro–Wilk test was used to determine the distribution of comfort data. Since *n* > 50 (Group1: handmade splint (*n* = 70); Group2: 3D-printed splint (*n* = 70)) we used the normal approximation to calculate the *p*-value. The *p*-value equaled $24.33\times 10^{-9}$ The test showed a significant difference from normality, W(140) = 0.9, *p* < 0.001. Skewness shape: asymmetrical, right/positive (pval = 0.005). Kurtosis shape: platykurtic, short, thin tails (pval = 0.037). The null hypothesis of the normal data distribution was rejected; thus, we had to use a non-parametric test and the independence assumption, which means that the data were not connected in any way. The observations between groups were independent; the groups were made up of different people. The measured variable was the subjective feeling of the proband [56,57]. At the same time, secondary phenomena affecting the evaluation were also considered (toothache with subsequent treatment, draft, weather-sensitive or other complications, etc.). The obtained values had zero correlation and a non-parametric distribution. Therefore, the Mann–Whitney U test was used for analysis. In statistics, it is a nonparametric test of the null hypothesis in which, for randomly selected values X and Y from two populations (treatment), the probability of X being more significant than Y is equal to the probability of Y being more significant than X.

Non-parametric Mann–Whitney U test (Wilcoxon rank-sum test) compares the probability of obtaining a higher value from handmade splint data with the possibility of receiving a higher value from a 3D-printed splint evaluation. Test calculation: two (H_1_: Splint1 ≠ Splint2). Sample size *n* = 140. Exact: false. The *p*-value equals 0.1475. Since *p*-value > α, H_0_ cannot be rejected. The difference between the randomly selected value of Splint1 and Splint2 data is not big enough to be statistically significant (Figure 7 and Figure 8).

## 4. Discussion

Establishing a correct diagnosis is the fundamental basis for effective treatment. It is important to diagnose the condition accurately before proceeding with any treatment plan. The human body is a wonderfully complex system, but we still have much to learn about how it works. Unfortunately, our current medical and technological abilities do not always allow us to repair, restore, or create certain physiological functions. It is important to remember that patients are individuals who deserve clear and simple solutions rather than being subjected to numerous examinations. Unfortunately, a series of examinations precede to determine the diagnosis of TMD. During a comprehensive dental analysis, it is of utmost importance to actively search for initial indications related to diseases of the orofacial area. Neglect of these signs may lead to permanent and severe damage to the dentition, along with joint dysfunction. During the diagnosis, it is essential to observe the characteristic manifestations and the signs causing difficulties for the patient. Occlusal parafunction includes bruxism (clenching, grinding) [27,58], lip biting, pain [1,4,59], and abnormal posturing of the jaw [1,9,60], as well as jaw clicking, muscle tenderness, pain during the opening, and limited mouth opening [7]. Tooth wear is mostly caused by bruxism [61]. It has been observed that the forces generated during an episode of bruxism are six times higher than the forces generated during regular mastication [61]. Bruxism, a chronic dental condition, is characterized by multifactorial etiology [1] and poses significant diagnostic and treatment challenges [62]. Despite extensive research, no definitive diagnostic or treatment protocol has been established as the gold standard [60,62]. Awake bruxism (AB) and sleep bruxism (SB) are two distinct conditions characterized by the involuntary clenching and grinding of teeth, respectively [26,58]. AB typically occurs during waking hours and is often linked to psychological factors, while SB takes place during sleep and is considered an oromandibular behavior that involves stereotyped movements of the jaw [27]. SB is characterized by phasic and/or tonic contractions of the masseter and other jaw muscles [29,63]. The clinical features for diagnosis of bruxism include complaints of jaw muscle discomfort, fatigue, stiffness, and/or occasional headaches, the presence of tooth wear, tooth sensitivity, muscle hypertrophy, temporomandibular joint clicking or jaw lock, and tongue indentation. The clinical diagnosis of bruxism is based on an orofacial examination and is usually supported by the patient’s history, self-reports, or parental/partner reports. Many patients with sleep bruxism are not aware of grinding [60]. The prevalence of SB in the adult population is estimated to be approximately 8–10% [58]. Pain management in the orofacial region is predicated based on the primary diagnosis.

Input data can be obtained through impressions, intraoral scanners, X-ray CT scans, or CBCT (cone-beam computed tomography) [64]. Medical examination, diagnosis, prognosis, or patient education before and after the procedure are entirely within the competence of doctors. In the future, there may be merit in contemplating electromyography and partnering it with neurologists. However, EMG biofeedback devices have some disadvantages, i.e., the EMG signals can be affected by electrode position, posture [10,62], and skin resistance [58]. It is also tricky for bruxers to tolerate the device while asleep as the electrodes are attached to the masseter and/or temporalis muscles [58]. Maintaining a meticulous balance between masticatory muscles’ functionality, the condyle–disc–articular socket’s anatomical relationship in the temporomandibular joint, and the interrelation of the upper and lower jaw teeth is paramount during treatment. This balance is critical, irrespective of the etiology of the underlying condition [1,16,31].

Without proper and timely orthodontic intervention, the condyle will gradually adapt to a new position, which may be pathological [65]. For TMD conservative treatment, an occlusal splint or physiotherapy is indicated. The type is assessed individually. One of the main challenges in temporomandibular rehabilitation [66,67] is accurately assessing the dynamic relationship between the occlusal surface and the condylar position [68]. A recent study compared the mechanical and computerized registration methods used with two selected kinematic face bows. It was found that mechanical facebow handling has a higher risk of hand-measuring errors in tracing. The significant difference in the measurements of the condylar path inclination is most likely a result of the differences in the assumptions of the registration techniques. The authors recommend using an articulator compatible with a facebow whose measurement has been completed. The authors assume that virtual tools for mandibular tracing with extremely high diagnostic potential will be introduced to daily practice soon [68]. In the case of the occlusal splint, identification of the centric relation of the mandible to the maxilla and, thus, the use of an anterior deprogrammer is necessary for the treatment of pain in the facial area caused by muscle disharmony and in all conditions where the vertical relationship of the upper and lower jaw changes due to damage to the teeth or partial or complete loss of teeth. Determining CR using available medical aids does not always meet the treatment requirements. Specifically, front deprogrammers face challenges related to the dimensions of the central incisors or the fixation in the modeling material. A collaboration between biomedical engineers, dental technics, and doctors from a dental clinic specializing in conservative dentistry and prosthetics was formed to address these requirements in dental practice. This research (case study) aims to improve the available medical aids for TMD treatment.

The task was to develop a complete data protocol of the production process, CAD design, and 3D printing of the front deprogrammers of the jaws. This is about applying digitization, CAD modeling, and additive manufacturing from biocompatible materials [44,46,48]. This study described the development and design of the anterior deprogrammer, a dental device used for neuromuscular adjustment that separates the posterior teeth to release the pterygoid muscle, allowing the condyles to settle into an optimal position. It focused on creating simple and practically applicable tools for the prescription (anterior deprogrammer in three sizes) and secondly for the communication between the dentist and the patient (computer application). This research dealt with designing, printing, and testing created medical aids in a home environment. The tested hypothesis was whether, according to these novel tools, it was possible to produce functional occlusal splints, which could be manufactured using any current technology. Another idea tested was whether manufactured occlusal splints differed in patients’ subjective perception of effectiveness and comfort. This research was conducted at a clinic specializing in restorative dentistry, which examines an average of 54 patients with TMD annually. Based on the statistical requirements, a minimum of 18 samples are needed to meet the necessary criteria (confidence level: 90%, margin of error: ±10%, population proportion: 10% [58]). This research was a pilot case study involving two probands who tested the level of the procedural algorithm and were willing to send feedback daily.

Three different sizes of anterior deprogrammers were designed based on extensive studies regarding the width of the central incisors [8,12,13,15,69]. For the tool to be satisfactory for determining the CR of the jaw and mandible, it must be appropriately designed. In the ideal case, it copies the course of the vestibular surfaces of the frontal teeth in the upper jaw, at least to the extent of the width of the central incisors; it may also include the lateral incisors. Its width should not extend into the canine area. The area simulating the anteroposterior movement of the lower jaw must be parallel to the chewing plane. It should be long enough to support the edges of the lower incisors throughout the movement. For example, the deprogrammer is fixed to the upper teeth using thermoplastic impression materials, ensuring sufficient tool stability on the upper dentition.

After a confirmatory CT examination, the dentist will utilize a dimensionally adequate anterior deprogrammer to induce a physiologically correct relation between the upper and lower jaw. This will result in a satisfactory response of the masticatory muscles. The dental technicians will use their material and professional experience to create occlusal splints based on the dentist’s prescription. This approach ensures that patients receive the highest level of dental care possible. The ideal design of the device should guide the patient comfortably into the position of the jaw in relation to the jaw, which is called the centric jaw relationship—the movement of the jaw in the anterior-posterior dimension should copy the occlusal splint. This is the definition which is used when describing the 3D orientation of the human dentition [50]. Unfortunately, it is difficult for “real bruxers” to tolerate the device well while asleep [58]. In general, splints that only touch the incisors are not indicated for TMDs as they act like dental braces and deform the occlusal plane. In summary, occlusal splints, including the anterior repositioning splint, hard stabilization splint, soft stabilization splint, mini-anterior splint, and prefabricated splint, may be considered more effective therapies for arthrogenous and myogenous temporomandibular disorders (TMDs) when compared to untreated control patients and non-occluding splints. This conclusion is based on the available evidence, which suggests that occlusal splints provide beneficial effects for the management of TMDs. Therefore, clinicians may be advised to consider the use of occlusal splints as a viable option for the treatment of TMDs, given their potential to improve patients’ symptoms and quality of life [70,71].

Dental technology has a role in terms of preventing and reducing tooth wear [61,72,73,74]. In the manufacturing of medical devices, the methods used play an essential role in achieving the desired properties [47,49,68,75]. Night guards have gained considerable popularity owing to their aesthetic appeal and flexibility in modern orthodontics. Traditional techniques have limitations [47], such as dimensional inaccuracies caused by thermoplastics’ conventional thermoforming, which leads to discomfort and treatment failure [47,68,73]. Heat-cured polymethyl methacrylate is a commonly used material in the production of occlusal splints, which are useful in treating functional disorders of the stomatognathic system. It possesses many beneficial properties, including adequate hardness, low shrinkage level, chemical stability, resistance to abrasion [40], and easy processing and handling. Its cost-effectiveness is also improved as it is cheaper than light-cured resin. However, it is not a perfect material as some individuals may experience hypersensitivity [76,77] to the monomer present in acrylate materials. This can lead to skin allergies and respiratory allergies in both dental technicians and patients. Additionally, acrylate materials have a strong, pungent odor that disappears after polymerization [77,78].

Additive manufacturing (AM) offers a high-precision alternative [49,52,54,79,80]. AM’s benefits, such as the capacity to produce complex objects, low waste, low energy consumption, and design flexibility, make it an attractive choice for creating heterogeneous multi-materials, functionally graded materials (FGMs), and homogenous single materials. This shift in preference can be attributed to the technological advancements in the field of 3D printing that have created high-quality and customized night guards [45,71].

Among the AM methods, photo-curing 3D printing is one of the most advanced methods, utilizing photosensitive resin as its raw material [49]. Photopolymer resins are materials used in various additive manufacturing processes, such as stereolithography or digital light processing, to create three-dimensional objects layer by layer.

The success of splint therapy is heavily reliant on the choice of material. Factors, such as biocompatibility, ease of fabrication [78], adjustability, cost [24], and patient preference must be considered during material selection [81,82]. Using bio-compatible and medically certified materials, 3D printing technology produces occlusal splints with high geometric accuracy [47,80,83,84]. Advances in material science and manufacturing techniques are giving rise to novel materials and techniques [44,45,46]. That is a major reason why there is a growing demand for 3D-printed biocompatible materials [78,83]. However, most of the evidence is based on in vitro studies with different methodologies, limiting their validity in daily practice [81,82]. Advancements in the understanding of applied biomechanics, alongside an increasing number of biomedical studies, have led to marked improvements in the reliability of orthodontic therapy. These developments are complemented by the evolution of thermoplastic materials, which have further contributed to the field’s progress [84]. In this research, a medically certified material was used. Dental LT Clear by Formlabs resin is a relatively new material, and its biomechanical resistance properties are still the subject of research. A previous biomechanical study [45] tested the Dental LT Clear by Formlabs material. This study evaluated 240 specimens in two shapes, dumbbell and rectangular, through compression and tensile tests. The results showed that polishing had a significant impact on the compression modulus values. The unpolished and non-aged specimens measured 0.87 ± 0.02, while the polished group measured 0.086 ± 0.03 after polishing. The results were significantly affected by artificial aging. The polished group measured 0.73 ± 0.05, while the unpolished group measured 0.73 ± 0.03. On the other hand, the tensile test revealed that the specimens showed the highest resistance when polishing was applied. Artificial aging affected the tensile test and reduced the force to damage the specimens. The tensile modulus had the highest value when applying polishing (3.00 ± 0.11) [44]. Based on these findings, the following conclusions were drawn: 1. Polishing did not alter the properties of the examined resin. 2. Artificial aging reduced resistance in both compression and tensile tests. 3. Polishing reduced the damage to the specimens in the aging process [45].

In addition, recent research has demonstrated that sterilization can yield benefits by reducing the elution of monomers while improving the microhardness of the resin. Specifically, autoclaving at a temperature of 132 °C for 4 min has been identified as particularly influential [45]. This is excellent news, precisely because of its benefits in the repeated use of anterior deprogrammers and during the prefabrication of occlusal splints.

Evaluating a patient’s condition is often predicated on their subjective perception of whether they are improving [1,60]. This can pose a challenge when verifying and objectifying the situation [85,86]. In cases of muscle problems, splints are commonly prescribed as a means of relaxation. In the future, there may be merit in contemplating electromyography and partnering with neurologists. It should be noted that, however, the results would still be subject to debate and could require further extensive research. Additionally, many patients are disinclined to undergo additional examinations.

This study also dealt with a patient evaluation of the treatment. A comfort scale was utilized, and participants could leave notes during the evaluation. For instance, one participant mentioned, “I caught a cold. That is why I could not tolerate the splint.” However, none of the notes had any impact on the rating. The negative feedback provided was the reason for seeking a medical consultation. There were setbacks during the research, a common problem in such studies. Establishing a long-term relationship with the proband was highly challenging, particularly given their painful condition. However, a simple application allowed the operator to monitor the patient’s conservative treatment and intervene quickly in re-education and motivation, which proved helpful [60,79]. Building a solid relationship between the patient and the doctor is crucial.

## 5. Conclusions

The anterior deprogrammer was specially designed based on the central incisors’ width. It enabled the creation of a prescription for creating an occlusal plate using traditional and digital workflows. The patients evaluated the success of the conservative treatment for ten weeks. For this purpose, a computer application was created for communication between the patient and the doctor. It proved to be crucial. Initial results are promising; no statistically significant difference was found between the productive technologies. A limitation of this study is its small sample size. Extensive research is needed to evaluate the effectiveness of TMD treatments.

## Figures and Tables

**Figure 1 bioengineering-10-01379-f001:**
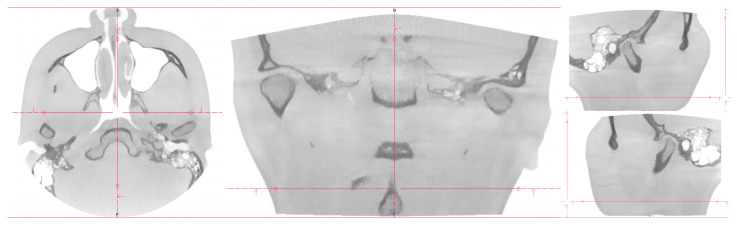
CT imaging of proband No. 1. The coronal, axial, and sagittal views.

**Figure 2 bioengineering-10-01379-f002:**
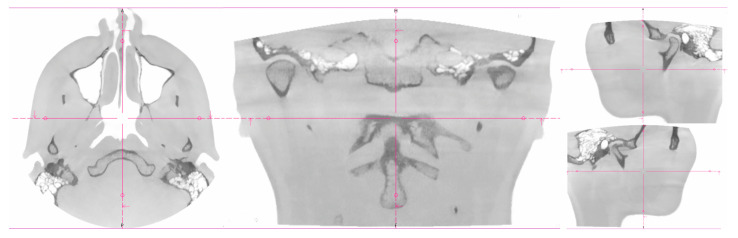
CT imaging of proband No 2. The Coronal, Axial, and Sagittal views.

**Figure 3 bioengineering-10-01379-f003:**
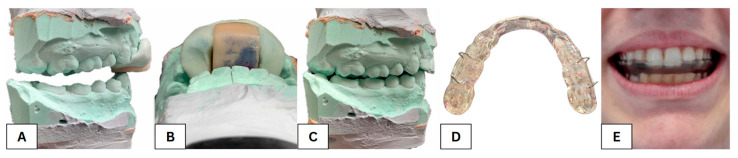
View of the plaster models of the upper and lower dentition fixed in a centric relationship with the sled deprogrammer (**A**), a detailed picture of the bite into the deprogrammer plate (**B**), and the modified relationship of the upper and lower dentition based on the centric relationship with the first contact in the area of the second molars (**C**), a traditionally made occlusal splint (**D**) and a traditionally made occlusal splint inside the mouth (**E**).

**Figure 4 bioengineering-10-01379-f004:**
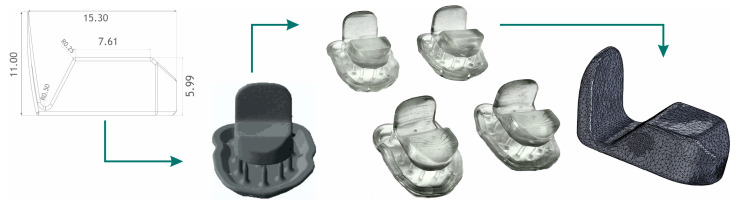
Anterior deprogrammer production algorithm according to the width of central incisors. CAD design, pre-processing, post-processing, and reverse dimensional control.

**Figure 5 bioengineering-10-01379-f005:**
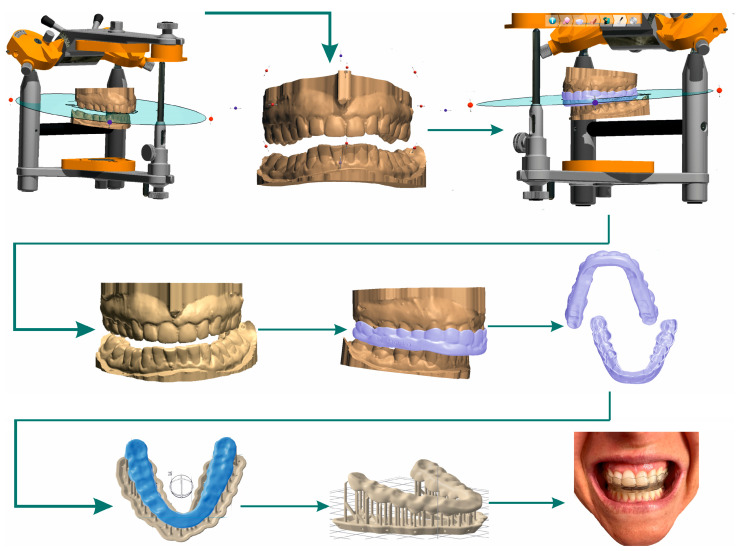
Digital workflow of occlusal splint CAD design, additive production algorithm, and fitted splint.

**Figure 6 bioengineering-10-01379-f006:**
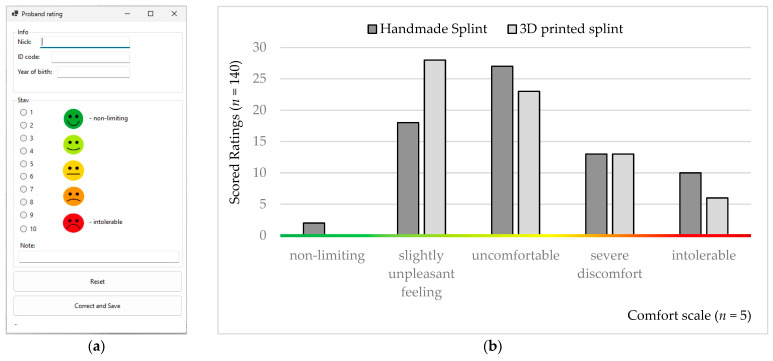
Rating: (**a**) visual demonstration from a computer application; (**b**) histograms from the ten-week comfort assessment obtained from the probands.

**Figure 7 bioengineering-10-01379-f007:**
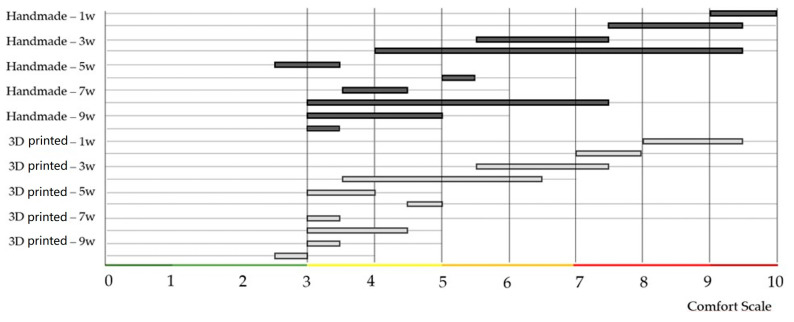
Evaluations of two different types of conservative treatment were visualized after individual weeks.

**Figure 8 bioengineering-10-01379-f008:**
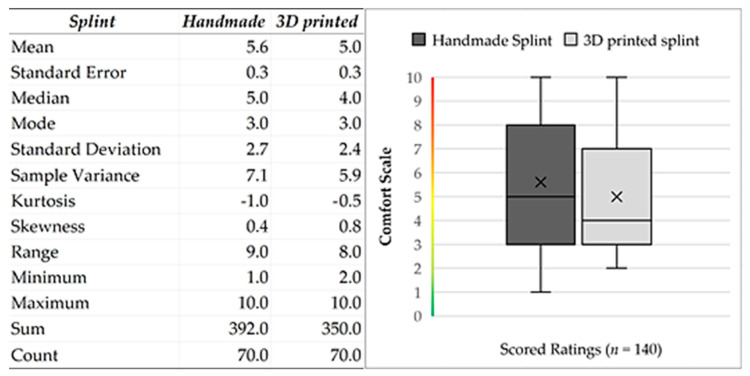
Descriptive statistics of input data. Descriptive statistics of input data. Graphical visualization of the dataset from patients based on the characteristics of the following five elements: the minimum, maximum, sample median, and first and third quartiles.

## Data Availability

Data will be provided upon a reasonable request.

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
