# Peer review of "Literature Review of an Anterior Deprogrammer to Determine the Centric Relation and Presentation of Cases"

_bioengineering, 2023, doi:10.3390/bioengineering10121379_

Round 1

Reviewer 1 Report

Comments and Suggestions for Authors

Abstract, it is not quite clear what is the main method used in this paper, also what is the main conclusion. These need to be concise and in a good logic.

Introduction is too short. Relevant literatures should be enriched, avoid group citations with large numbers of references, such as 1-7, 9–15, 17-22.

The novelty of this paper is not quite clear. ‘Based on the above, our primary goal is to create a procedural algorithm for produing and testing the anterior deprogrammer and an additively made occlusal splint to alleviate the difficulties associated with bruxism and pain in the temporomandibular joint [23]’’ this sentence needs to be rewritten as it is too lengthy, also, this paragraph needs to be elaborated to emphasize the novelty of this paper. What is ref 23 for? Specifically?

Relevant literatures about 3D printing in biomedical should be introduced, such as ‘Characterization of Alginate–Gelatin–Cholesteryl Ester Liquid Crystals Bioinks for Extrusion Bioprinting of Tissue Engineering Scaffolds’, ‘A 3D-Printed Biomaterial Scaffold Reinforced with Inorganic Fillers for Bone Tissue Engineering: In Vitro Assessment and In Vivo Animal Studies’.

Discuss a bit more about the advantages and disadvantages of the two different types of conservative treatment.

Discuss a bit more about future development trend, and future work.

Author Response

Dear Reviewer,

Firstly, we would like to express our gratitude for your time, valuable advice, and insights, which have significantly improved our manuscript. Thank you very much!

  • We tried to revise the abstract following your instructions.

"The biomedical research aimed to create a beneficial, ecological, and readily available anterior deprogrammer to determine the centric relation (CR). This medical device is additively manufactured from a biocompatible material. Size is customizable based on the width of the patient's anterior central incisors. It is a pilot study, with two subjects participating in the case study. The task was to develop a complete data protocol for the production process, CAD design, and 3D printing of the anterior deprogrammers. The research focused on creating simple and practically applicable tools for the dentist's prescription (anterior deprogrammer in three sizes) and secondly for communication between the dentist and the patient (computer application). The tested hypothesis was whether, according to these novel tools, it is possible to produce functional occlusal splints, which could be manufactured using current technologies. The study compared a traditional splint with a digitally designed and 3D-printed one. The tested hypothesis was whether manufactured occlusal splints differ in patients’ subjective perception of comfort."

  • In our writing, we have consciously tried to expand the introduction and literature sections while avoiding group references. We apologize for incorrectly citing a reference. We have fixed the mistake and modified the text.  
  • We added information from the relevant literature you recommended to the manuscript and subsequently quoted it correctly.

Abdulmaged, A.I.; Soon, C.F.; Talip, B.A.; Zamhuri, S.A.A.; Mostafa, S.A.; Zhou, W. Characterization of Alginate–Gelatin–Cholesteryl Ester Liquid Crystals Bioinks for Extrusion Bioprinting of Tissue Engineering Scaffolds. Polymers 2022, 14, 1021. https://doi.org/10.3390/polym14051021.

Sithole, M.N.; Kumar, P.; Du Toit, L.C.; Erlwanger, K.H.; Ubanako, P.N.; Choonara, Y.E. A 3D-Printed Biomaterial Scaffold Reinforced with Inorganic Fillers for Bone Tissue Engineering: In Vitro Assessment and In Vivo Animal Studies. Int. J. Mol. Sci. 2023, 24, 7611. https://doi.org/10.3390/ijms24087611

  • The discussion is expanded on the advantages and disadvantages of two different types of conservative treatment future development trends, and future work.

Production technology... "In the manufacturing of medical devices, the methods used play an essential role in achieving the desired properties. In modern orthodontics, clear night guards have gained considerable popularity owing to their aesthetic appeal and flexibility. While traditional techniques have limitations, Additive Manufacturing (AM) offers a high-precision alternative"...

Future trends in prescription with kinematic face bows... Prescription..."virtual tools for mandibular tracing with extremely high diagnostic potential will soon be introduced to daily practice". 

Diagnosis Establishing..."In the future, there may be merit in contemplating electromyography and partnering with neurologists. However, EMG biofeedback devices have some disadvantages, i.e., the EMG signals can be affected by electrode position, posture, and skin resistance." 

Reviewer 2 Report

Comments and Suggestions for Authors

The paper offers valuable insights into the application of stereolithographic 3D printing in the dental and healthcare fields, particularly in the development and design of the Lucia jig for neuromuscular adjustment. This showcases the immense potential of additive manufacturing for producing dental devices tailored to specific functionalities.

§  The abstract is incompletely written and needs minor revision. The steps of the article should be mentioned first, and then the results should be presented quantitatively and qualitatively. Finally, the most important achievements should be mentioned.

§  Can the authors provide more information on the specific biocompatible dental materials used in the Formlabs Form 3B printer?

§  At first glance, the introduction is too short and lacks quantitative results. Also, the first paragraphs presented are primarily general and general information. At the end of the introduction, a suitable summary of the importance of the present issue should be provided. Also, discontinuity between paragraphs is evident in most of the introduction. It is suggested to rewrite the introduction.

§  Use the following resources to deepen the introduction and discussion. Shape memory performance assessment of FDM 3D printed PLA-TPU composites by Box-Behnken response surface methodology. 4D printing of PLA-TPU blends: effect of PLA concentration, loading mode, and programming temperature on the shape memory effect. A novel practical method for the production of Functionally Graded Materials by varying exposure time via photo-curing 3D printing.

§  What steps were taken in the study to consider any factors or variables that could have affected the patients' personal evaluation of the comfort of the splints?

§  Please provide more information about the computer application used for communication between patients, doctors, and operators. How did it simplify communication and enhance data collection and analysis?

§  Page 6, paragraph one, can you elaborate on the non-parametric test used and the assumption of independence?

§  What are the ideal design characteristics of the deprogrammer, in terms of its resemblance to the front teeth and its accommodation of the lower jaw's movement?

§  Page 8, what design considerations are important for an accurate jaw deprogrammer?

Comments on the Quality of English Language

***

Author Response

Dear Reviewer,

Firstly, we would like to express our gratitude for your time and valuable advice and insights, which have significantly improved our manuscript. Thank you very much!

  • We tried to revise the abstract following your instructions.

"The biomedical research aimed to create a beneficial, ecological, and readily available anterior deprogrammer to determine the centric relation (CR). This medical device is additively manufactured from a biocompatible material. Size is customizable based on the width of the patient's anterior central incisors. It is a pilot study, with two subjects participating in the case study. The task was to develop a complete data protocol for the production process, CAD design, and 3D printing of the anterior deprogrammers. The research focused on creating simple and practically applicable tools for the dentist's prescription (anterior deprogrammer in three sizes) and secondly for communication between the dentist and the patient (computer application). The tested hypothesis was whether, according to these novel tools, it is possible to produce functional occlusal splints, which could be manufactured using current technologies. The study compared a traditional splint with a digitally designed and 3D-printed one. The tested hypothesis was whether manufactured occlusal splints differ in patients’ subjective perception of comfort."

  • We add more information on the specific biocompatible dental materials used in the Formlabs Form 3B printer.

2.2. Materials

„The selected Photopolymer Resin Dental LT Clear is ideal for rigid splints, occlusal guards, and other direct-printed long-term orthodontic appliances because it is a Class IIa long-term biocompatible resin. It is a material suitable for use over an extended pe-riod on mucosal and skin surfaces. Dental materials must be biocompatible, especially when they will be in close contact with tissues for extended periods of time. [24, 25] Technical testing is utilized to evaluate a material's biocompatibility and guarantee that it is safe for human usage. [26] Dental LT Clear is tested at NAMSA, Chasse sur Rhône in France, and is certified biocompatible per EN-ISO 10993-1:2009/AC:2010 The product follows ISO Standards: EEN ISO 1641:2009; EN-ISO 10993-1:2009/AC:2010; EN-ISO 10993-3:2009; EN-ISO 10993-5:2009; EN 908:2008. [27, 28, 29]This resin has a high translucency and a nonlinear compression resistance of up to 600 N, which is equivalent to the force of biting. [26] Teeth are the most resilient hu-man tissues. [30] The 3D-printed splints, therefore, need post-curing [31] required. Second, foods and drinks can impact the characteristics of intraoral appliances, re-ducing their aesthetic appeal and jeopardizing their structural integrity. Conditions within the mouth can also affect a material's characteristics.“

  • In our writing, we have consciously tried to expand the introduction, and a suitable summary of the importance of the present issue is provided.

...“the primary goal is to design and manufacture a modifiable anterior deprogrammer. Considering ecological principles, an additive production method based on CAD data of a 3D-designed model [16] was used as a production technology. The tested hypothesis was whether, according to this medical device, it is possible to produce functional occlusal splints, which could be manufactured using any current technology. Another hypothesis tested was whether manufactured occlusal splints differ in patients’ subjective perception of effectiveness and comfort. For this purpose, it was necessary to create a new procedural algorithm considering the multifactorial nature of TMD to monitor the patient's subjective perception of comfort. Based on previous research, it can be proven that temporomandibular joint pain goes beyond the physiology of articulation and is significantly influenced by the external and internal influences of the human body environment.“

  • We added information from the relevant literature you recommended to the manuscript and subsequently quoted it correctly.

Rahmatabadi, D., Soltanmohammadi, K., Pahlavani, M. et al. Shape memory performance assessment of FDM 3D printed PLA-TPU composites by Box-Behnken response surface methodology. Int J Adv Manuf Technol 127, 935–950 (2023). https://doi.org/10.1007/s00170-023-11571-2

Bazyar M., M; Bozorgnia, Tabary, S., A., A.; Rahmatabdi, D.; Mohammadi, K.; Hashemi, R. A novel practical method for the production of Functionally Graded Materials by varying exposure time via photo-curing 3D printing: Journal of Manufacturing: Processes, vol. 103, 2023,136-143, ISSN 1526-6125

Rahmatabadi, D., Ghasemi, I., Baniassadi, M. et al. 4D printing of PLA-TPU blends: effect of PLA concentration, loading mode, and programming temperature on the shape memory effect. J Mater Sci 58, 7227–7243 (2023). https://doi.org/10.1007/s10853-023-08460-0

  • Your question: "What steps were taken in the study to consider any factors or variables that could have affected the patients' personal evaluation of the comfort of the splints?"

Answer in Discussion in Assessment of Conservative Treatment.: „Evaluating a patient's condition is often predicated on their subjective perception of whether they are improving. This can pose a challenge when verifying and objectifying the situation. In cases of muscle problems, splints are commonly prescribed as a means of relaxation. In the future, there may be merit in contemplating electromyography and partnering with neurologists. It should be noted, however, that the results would still be subject to debate and could require further extensive research. Additionally, many patients are disinclined to undergo additional examinations.

During the evaluation process, a comfort scale was utilized, and participants could leave a note. For instance, one participant mentioned, "I caught a cold. That's why I could not tolerate the splint." However, none of the notes had any impact on the rating. The negative feedback provided was the reason for seeking medical consultation.“

  • Your question: Please provide more information about the computer application used for communication between patients, doctors, and operators. How did it simplify communication and enhance data collection and analysis?

2.3.3. Procedural algorithm of occlusal splint testing

„The application is created in the Visual Studio environment in the C# programming language. It is an application based on Windows Form Application with a user interface. It contains an evaluation part, a communication part, and a backup part and uses server storage to store communication data between the patient, the doctor, and the operator. A firewall and general protection protect the data, as the server is located on the university network and is securely stored on physical storage without the possibility of cloud backup. The application has versatile use in testing medical devices and is the property of the Department of Biomedical Engineering and Measurement.“

  • Your question: Page 6, paragraph one, can you elaborate on the non-parametric test used and the assumption of independence?
  1. Results

„It was used the Mann-Whitney U test for analysis. In statistics, it is a nonparametric test of the null hypothesis that, for randomly selected values X and Y from two populations (treatment), the probability of X being more significant than Y is equal to the probability of Y being more significant than X.

The independence assumption, which means, the data isn't connected in any way. The observations between groups were independent; the groups were made up of different people. The measured variable is the subjective feeling of the proband. [32] At the same time, secondary phenomena affecting the evaluation must also be considered (tooth-ache with subsequent treatment, draft, weather-sensitive or other complications, etc.). The obtained values have zero correlation and non-parametric distribution.“

  • Your question: What are the ideal design characteristics of the deprogrammer, in terms of its resemblance to the front teeth and its accommodation of the lower jaw's movement?

Answer in Discussion in Occlusal Splint Production.

„The ideal design of the device should guide the patient comfortably into the position of the jaw in relation to the jaw, which is called the centric jaw relationship – the move-ment of the jaw in the anterior-posterior dimension should copy the occlusal splint– this is the definition, which is used when describing the 3D orientation of the human dentition.“

  • Page 8, what design considerations are important for an accurate jaw deprogrammer?

Answer in Discussion in Prescription.

...“anterior deprogrammers face challenges related to the dimensions of the central incisors or fixation in the modeling material. To address these requirements in dental practice, a collaboration between biomedical engineers, dental technics, and doctors from a dental clinic specializing in conservative dentistry and prosthetics was formed. This research aims to improve the available medical aids for TMD treatment.“

Reviewer 3 Report

Comments and Suggestions for Authors

This article needs a lot of improvements and not suitable to publication in this form.

The abstract needs background. Plus, a lot of abbreviations were made without any full name before.

The introduction is not suitable in this form each long paragraph with 7 references no hypothesis

Please avoid the use of we and our in text.

Add sample size calculation

The discussion is too small and the conclusion is tooo long. It is the first time that i saw such an article with such a poor presentation.

Comments on the Quality of English Language

Moderate

Author Response

Dear Reviewer,

Firstly, we would like to express our gratitude for your time, valuable advice, and insights, which have significantly improved our manuscript. Thank you very much!

  • We tried to revise the manuscripts following your instructions.
  • We tried to revise the abstract and Introduction following your instructions.

Abstract: The increasing demand for dental aesthetics, articulation corrections, and solutions for pain and frequent bruxism requests quick and effective restorative dental management. The bio-medical research aimed to create a beneficial, ecological, and readily available anterior depro-grammer to determine the centric relation (CR). This medical device is additively manufactured from a biocompatible material. Size is customizable based on the width of the patient's anterior central incisors. It is a pilot study, with two subjects participating in the case study. The task was to develop a complete data protocol for the production process, CAD design, and 3D printing of the anterior deprogrammers. The research focused on creating simple and practically applicable tools for the dentist's prescription (anterior deprogrammer in three sizes) and secondly for communica-tion between the dentist and the patient (computer application). The tested hypothesis was whether, according to these novel tools, it is possible to produce functional occlusal splints, which could be manufactured using current technologies. The study compared a traditional splint with a digitally designed and 3D-printed one. The tested hypothesis was whether manufactured occlusal splints differ in patients’ subjective perception of comfort. Each conservative treatment was mon-itored over ten weeks. Initial results are promising; no statistically significant difference was found between the productive technologies.

Introduction

The rising trend for dental and maxillofacial aesthetics, even in the case of tem-poromandibular disease (TMD) [1, 2], creates enormous pressure on dental clinics and dental technicians. The worldwide effort for an ecological approach seeks solutions by reducing the carbon footprint and production waste. In this context, mnemonic and digital visual aids are coming to the fore, facilitating the understanding of the broader context of the human body and its interaction with the action of internal and external forces [2].

Based on these sought-after quantities, biomedical research was created regarding anterior deprogrammers for the upper and lower jaw designed to solve the unbalanced function of the temporomandibular joint (TMJ) [3, 4] and masticatory muscles. Tem-poromandibular disorders (TMD) involve several factors or causes [1], representing a heterogeneous group of conditions concerning TMJ and the surrounding structures [2, 5, 6]. Symptoms include jaw clicking, muscle tenderness, pain during the opening, and limited mouth opening [1, 7]. The disease etiology is multifactorial and is made up of biological factors [8] (e.g., internal derangements in TMJ), psychological factors (e.g., depression, anxiety, and stress), and social factors (e.g., a learned response to pain). Psychological influences are considered highly significant in the development of TMD [1, 2]. The results of some studies’ qualitative analyses have shown a positive associa-tion between migraines and TMDs (general) and an absence of an association between articular TMD without pain [2, 4, 8] (disk displacement and degenerative diseases) and primary headaches. The Meta-analysis results confirmed a positive association be-tween both pain-related TMD and migraine and chronic headaches. The association between pain-related TMD and Tension-type headaches is controversial [4].

Many current studies [5, 6] consider TMD to be associated with an imbalance of the whole body, and others [3, 7, 8] deal with the impact of TMD on the quality of a human's sleep. Current research focuses on a conservative solution through a night occlusal splint [8, 9, 10, 11] and considers it a successful treatment, as well as physio-therapy. Occlusal splints, known as night guards, orthotics devices, oral appliances, or modified nociceptive trigeminal inhibition splints [12], are used routinely in dental management [12, 13]. Occlusal splints help reduce tension, decrease muscle activity, and prevent harmful effects caused by bruxism and TMDs [13]. It is a tool to treat pa-tients with TMD and bruxism for occlusal stabilization and to reduce dentition wear [13]. Current conservative treatment options for TMDs include patient education and motivation to use occlusal splint therapy or physiotherapy [13, 14].

The biomechanical influence of the device. To define the geometric dimensions of the night guard, however, a unique tool is needed to determine the centric relation. However, the available medical aids do not always meet the treatment requirements [13, 14]. The discussion centers on the requirements imposed upon front deprogram-mers in relation to the dimensions associated with the width of central incisors as well as the fixation in modeling material. It is essential to ensure that front deprogrammers adhere to these requirements to guarantee an accurate and effective outcome. 

Based on the requirements of the dental practice, research was created that com-bines biomedical engineers and dental technics collaborating with doctors from a den-tal clinic specializing in conservative dentistry and prosthetics. This joint activity fo-cuses primarily on setting the centric relation (CR) of the upper jaw (maxilla) and the lower jaw (mandible). The so-called centric relation defines the balance. According to Dawson, regardless of vertical dimension or tooth position, CR is the connection be-tween the mandible and maxilla when the condyle-disk assemblies are correctly aligned and in the most superior position against the eminent [12, 15]. The con-dyle-disk assemblies are reinforced medially in their highest position. Centric relation is, therefore, also in the middle [12]. Condyle-disk assemblies that are centrally aligned can withstand the maximum loading applied by the elevator muscles without experi-encing any pain [16].

The research effort is to find a simple and effective solution applicable in practice and looking for the right balance between the function of the masticatory muscles, the anatomical relationship of the condyle-disc-articular socket in the temporomandibular joint, and the relationship between the teeth of the upper and lower dentition, regard-less of the etiology of the disease [1].

Based on the above, the primary goal is to design and manufacture a modifiable anterior deprogrammer. Considering ecological principles, an additive production method based on CAD data of a 3D-designed model [17] was used as a production technology. The tested hypothesis was whether, according to this medical device, it is possible to produce functional occlusal splints, which could be manufactured using any current technology. Another hypothesis tested was whether manufactured occlu-sal splints differ in patients’ subjective perception of effectiveness and comfort. For this purpose, it was necessary to create a new procedural algorithm considering the multifactorial nature of TMD to monitor the patient's subjective perception of comfort. Based on previous research, it can be proven that temporomandibular joint pain goes beyond the physiology of articulation and is significantly influenced by the external and internal influences of the human body environment.

  • Please avoid the use of we and our in text.

We have rewritten the sentences in the passive voice

  • Add sample size calculation

Answer in „3. Results“

„The sample size is calculated based on the statistical requirements for a popula-tion size of 54 patients (Sample size: n = 18, Confidence Level: 90%, Margin of Error: ±10%, Population Proportion: 10% [27]). The submitted research is a pilot case study involving two probands who tested the level of the procedural algorithm and were willing to send feedback daily.“

The research was conducted at a clinic that specializes in restorative dentistry, which examines an average of 54 patients with TMD annually. The prevalence of SB in the adult population is estimated to be approximately 8–10%.

  • The discussion is too small and the conclusion is tooo long. It is the first time that i saw such an article with such a poor presentation.

We apologize for all the shortcomings; we tried to correct the mistakes.

Discussion

Diagnosis Establishing. The human body is a wonderfully complex system, but we still have much to learn about how it works. Unfortunately, our current medical and technological abilities do not always allow us to repair, restore, or create certain physiological functions. It is important to remember that patients are individuals who deserve clear and simple solutions rather than being subjected to numerous examinations. Unfortunately, a series of examinations precede to determine the diagnosis of TMD. During a comprehensive dental analysis, it is of utmost importance to actively search for initial indications related to diseases of the orofacial area. Neglect of these signs may lead to permanent and severe damage to the dentition, along with joint dysfunction. Pain management in the orofacial region is predicated on the primary diagnosis. Input data can be obtained through impressions, intraoral scanners, X-ray CT, or CBCT (Cone Beam Computed Tomography) [28]. Medical examination, diagnosis, prognosis, or patient education before and after the procedure are entirely within the competence of doctors. In the future, there may be merit in contemplating electro-myography and partnering with neurologists. However, EMG biofeedback devices have some disadvantages, i.e., the EMG signals can be affected by electrode position, posture, and skin resistance [27,29]. It is also tricky for bruxers to tolerate the device while asleep with the electrodes attached to the masseter and/or temporalis muscles [27]. Maintaining a meticulous balance between masticatory muscles' functionality, the condyle-disc-articular socket's anatomical relationship in the temporomandibular joint, and the interrelation of the upper and lower jaw teeth is paramount during treatment. This balance is critical, irrespective of the etiology of the underlying condition [1,14,30].

Signs. Occlusal parafunction includes bruxism (clenching, grinding) [27], lip biting, pain [1,4], and abnormal posturing of the jaw [31] and jaw clicking, muscle tenderness, pain during the opening, and limited mouth opening [7]. Bruxism, a chronic dental condition, is characterized by multifactorial etiology [1] and poses significant diagnostic and treatment challenges [29]. Despite extensive research, no definitive diagnostic or treatment protocol has been established as the gold standard [29,31]. Awake bruxism (AB) and sleep bruxism (SB) are two distinct conditions characterized by the involuntary clenching and grinding of teeth, respectively [27]. AB typically occurs during waking hours and is often linked to psychological factors, while SB takes place during sleep and is considered an oromandibular behavior that involves stereotyped movements of the jaw. SB is characterized by phasic and/or tonic contractions of the masseter and other jaw muscles [32]. The clinical features for diagnosis of bruxism include complaints of jaw muscle discomfort, fatigue, stiffness, and/or occasional headaches, the presence of tooth wear, tooth sensitivity, muscle hypertrophy, temporomandibular joint clicking or jaw lock, and tongue indentation. The clinical diagnosis of bruxism is based on orofacial examination and is usually supported by patient history, self-reports, or parental/partner reports. Many sleep bruxism patients are not aware of grinding [31]. The prevalence of SB in the adult population is estimated to be approximately 8–10% [27].

Prescription. For TMD conservative treatment, an occlusal splint or physiotherapy is indicated. The type is assessed individually. One of the main challenges in temporomandibular rehabilitation is accurately assessing the dynamic relationship between the occlusal surface and the condylar position [33]. A recent study compared the mechanical and computerized registration methods used by the two selected kinematic face bows. It found mechanical facebow handling has a higher risk of hand-measuring errors in tracing. The significant difference in the measurements of the condylar path inclination is most likely a result of the differences in the registration techniques as-sumptions. The authors recommend using an articulator compatible with a facebow whose measurement has been done. The authors assume that virtual tools for mandibular tracing with extremely high diagnostic potential will be introduced to daily practice soon [33]. In the case of the occlusal splint, identification of the centric relation of the mandible to the maxilla and, thus, the use of an anterior deprogrammer is necessary for the treatment of pain in the facial area caused by muscle disharmony and in all conditions where the vertical relationship of the upper and lower jaw changes due to damage to the teeth or partial or complete loss of teeth. Determining CR using available medical aids does not always meet the treatment requirements. Specifically, front deprogrammers face challenges related to the dimensions of the central incisors or fixation in the modeling material. To address these requirements in dental practice, a collaboration between biomedical engineers, dental technics, and doctors from a dental clinic specializing in conservative dentistry and prosthetics was formed. This research aims to improve the available medical aids for TMD treatment.

Case Study. Pilot study. The task was to develop a complete data protocol of the production process, CAD design, and 3D printing of the front deprogrammers of the jaws. It is about applying digitization, CAD modeling, and additive manufacturing from biocompatible materials [20,23,24]. The study describes the development and design of the anterior deprogrammer, a dental device used for neuromuscular adjustment that separates the posterior teeth to release the pterygoid muscle, allowing the condyles to settle into an optimal position. It focused on creating simple and practically applicable tools for the prescription (anterior deprogrammer in three sizes) and secondly for communication between the dentist and the patient (computer application). The research deals with designing, printing, and testing created medical aids in a home environment. The tested hypothesis was whether, according to these novel tools, it is possible to produce functional occlusal splints, which could be manufactured using any current technology. Another idea tested was whether manufactured occlusal splints differ in patients’ subjective perception of effectiveness and comfort. The research was conducted at a clinic that specializes in restorative dentistry, which examines an average of 54 patients with TMD annually. Based on the statistical requirements, a minimum of 18 samples are needed to meet the necessary criteria (Confidence Level: 90%, Margin of Error: ±10%, Population Proportion: 10% [27]). The submitted research is a pilot case study involving two probands who tested the level of the procedural algorithm and were willing to send feedback daily.

Three different sizes of anterior deprogrammers have been designed based on extensive studies regarding the width of the central incisors [8,10,11,13,34]. For the tool to be satisfactory for determining the CR of the jaw and mandible, it must be appropriately designed. In the ideal case, it copies the course of the vestibular surfaces of the frontal teeth in the upper jaw, at least to the extent of the width of the central incisors; it may also include the lateral incisors. Its width should not extend into the canine ar-ea. The area simulating the anteroposterior movement of the lower jaw must be parallel to the chewing plane. It should be long enough to support the edges of the lower incisors throughout the movement. For example, the deprogrammer is fixed to the upper teeth using thermoplastic impression materials, ensuring sufficient tool stability on the upper dentition.

Following on a confirmatory CT examination, the dentist will utilize a dimensionally adequate anterior deprogrammer to induce a physiologically correct relation between the upper and the lower jaw. This will result in a satisfactory response of the masticatory muscles. The dental technicians, using their material and professional experience, will then create occlusal splints based on the dentist’s prescription. This approach ensures that patients receive the highest level of dental care possible.

Occlusal Splint Production. The ideal design of the device should guide the patient comfortably into the position of the jaw in relation to the jaw, which is called the centric jaw relationship – the movement of the jaw in the anterior-posterior dimension should copy the occlusal splint– this is the definition, which is used when describing the 3D orientation of the human dentition [35]. Unfortunately, it is difficult for “real bruxers” to tolerate the device well while asleep [27]. In general, splints that only touch the incisors are not indicated for TMD diseases as they act like dental braces and deform the occlusal plane.

Production technology. In the manufacturing of medical devices, the methods used play an essential role in achieving the desired properties [25,33,36]. In modern orthodontics, clear night guards have gained considerable popularity owing to their aesthetic appeal and flexibility. While traditional techniques have limitations [33], Additive Manufacturing (AM) offers a high-precision alternative [25,37].

There is a growing preference for 3D-printed materials due to their superior accuracy and precision [38]. This shift in preference can be attributed to the technological advancements in the field of 3D printing that have created high-quality and customized clear aligners [20]. AM's benefits, such as the capacity to produce complex objects, low waste, low energy consumption, and design flexibility, make it an attractive choice for creating heterogeneous multi-material, Functionally Graded Materials (FGMs), and homogenous single materials. Among the AM methods, photo-curing 3D printing is one of the most advanced, utilizing photosensitive resin as a raw material [25]. Photopolymer resins are materials used in various additive manufacturing processes, such as stereolithography or digital light processing, to create three-dimensional objects layer by layer.

Materials. Advancements in the understanding of applied biomechanics, alongside an increasing number of biomedical studies, have led to marked improvements in the reliability of orthodontic therapy. These developments are complemented by the evolution of thermoplastic materials, which have further contributed to the field’s progress [39]. In this research, it was used medically certified material. Dental LT Clear by Formlabs resin is a relatively new material and its biomechanical resistance properties are still the subject of research. The previous biomechanical study [20] was conducted to test the Dental LT Clear by Formlabs material. The study evaluated 240 specimens in two shapes, dumbbell and rectangular, through compression and tensile tests. The results showed that polishing had a significant impact on the compression modulus values. The unpolished and non-aged specimens measured 0.87 ± 0.02, while the polished group measured 0.086 ± 0.03 after polishing. The results were significantly affected by artificial aging. The polished group measured 0.73 ± 0.05, while the unpolished group measured 0.73 ± 0.03. On the other hand, the tensile test revealed that the specimens showed the highest resistance when polishing was applied. Artificial aging affected the tensile test and reduced the force to damage the specimens. The tensile modulus had the highest value when applying polishing (3.00 ± 0.11) [20]. Based on these findings, the following conclusions were drawn: 1. Polishing did not alter the properties of the examined resin. 2. Artificial aging reduced resistance in both compression and tensile tests. 3. Polishing reduced the damage to the specimens in the aging process [20].

In addition, recent research has demonstrated that sterilization can yield benefits by reducing the elution of monomers while improving the microhardness of the resin. Specifically, autoclaving at a temperature of 132°C for 4 minutes has been identified as particularly influential [20]. This is excellent news precisely because of the repeated use of anterior deprogrammers and during the prefabrication of occlusal splints.

Assessment of Conservative Treatment. Evaluating a patient's condition is often predicated on their subjective perception of whether they are improving [1,31]. This can pose a challenge when verifying and objectifying the situation. In cases of muscle problems, splints are commonly prescribed as a means of relaxation. In the future, there may be merit in contemplating electromyography and partnering with neurologists. It should be noted, however, that the results would still be subject to debate and could require further extensive research. Additionally, many patients are disinclined to undergo additional examinations.

During the evaluation process, a comfort scale was utilized, and participants could leave a note. For instance, one participant mentioned, "I caught a cold. That's why I could not tolerate the splint." However, none of the notes had any impact on the rating. The negative feedback provided was the reason for seeking medical consultation.

Critical analysis of the results

There were during the research, which is a common problem in such studies. Establishing a long-term relationship with the proband was extremely challenging, particularly given their painful condition. However, a simple application allowed the operator to monitor the patient's conservative treatment and intervene quickly in reeducation and motivation, which proved helpful [31,37]. Building a solid relationship between the patient and the doctor is crucial.

Reviewer 4 Report

Comments and Suggestions for Authors

Dear Authors, thank you for this interesting article. I would add some suggestions how to improve it though:

1. The TMD is a multifactorial problem, not only effecting from improper occlusion or muscular problems, eg. 

Topaloglu-Ak A, Kurtulmus H, Basa S, Sabuncuoglu O. Can sleeping habits be associated with sleep bruxism, temporomandibular disorders and dental caries among children? Dent Med Probl. 2022;59(4):517–522. doi:10.17219/dmp/150615

Florjański W, Orzeszek S. Role of mental state in temporomandibular disorders: A review of the literature. Dent Med Probl. 2021;58(1):127–133. doi:10.17219/dmp/132978

Please, note that in your introduction. Please, refer also to Okeson, that is a worldwide known professor that wrote a lot about that.

2. Fig. 3 - please add more photos, so that everyone could easily see the construction of the splint

3. In the discussion:

- add the aspect of "real bruxers" - the use of splint that touches only incisors could be a very unpleasent for them

- refer to artificial aging of materials that could be used to manufacture splints, eg Dental LT Clear,. Paradowska-Stolarz, A.; Wezgowiec, J.; Malysa, A.; Wieckiewicz, M. Effects of Polishing and Artificial Aging on Mechanical Properties of Dental LT Clear® Resin. J. Funct. Biomater. 202314, 295. https://doi.org/10.3390/jfb14060295

Could this influence the long-term quality of the splint and its precission?

4. In the discussion, please add an important limitation, which is a limited possibility to use articulators when manufacturing any occlusal splint and how important that is - refer to commonly used facebows (eg. Wieckiewicz M, Zietek M, Nowakowska D, Wieckiewicz W. Comparison of selected kinematic facebows applied to mandibular tracing. Biomed Res Int. 2014;2014:818694. doi: 10.1155/2014/818694.) and 3D systems (eg. ModJaw - refer to high costs eg.)

5. I would add in the title, that the article is a literature review and presentation of the cases, because that is indeed what it is.

Thank you.

Author Response

Dear Reviewer,

Firstly, we would like to express our gratitude for your time, valuable advice, and insights, which have significantly improved our manuscript. Thank you very much!

  • We added information about TMD from the relevant literature you recommended to the manuscript. In our writing, we have consciously tried to expand the introduction.

Topaloglu-Ak A, Kurtulmus H, Basa S, Sabuncuoglu O. Can sleeping habits be associated with sleep bruxism, temporomandibular disorders and dental caries among children? Dent Med Probl. 2022;59(4):517–522. doi:10.17219/dmp/150615

Florjański W, Orzeszek S. Role of mental state in temporomandibular disorders: A review of the literature. Dent Med Probl. 2021;58(1):127–133. doi:10.17219/dmp/132978

Okeson J, Porto FB, Furquim BD, Feu D, Sato F, Cardinal L. An interview with Jeffrey Okeson. Dental Press J Orthod. 2018 Nov-Dec;23(6):30-39. doi: 10.1590/2177-6709.23.6.030-039.int. PMID: 30672983; PMCID: PMC6340200

Réus JC, Polmann H, Souza BDM, Flores-Mir C, Gonçalves DAG, de Queiroz LP, Okeson J, De Luca Canto G. Association between primary headaches and temporomandibular disorders: A systematic review and meta-analysis. J Am Dent Assoc. 2022 Feb;153(2):120-131.e6. doi: 10.1016/j.adaj.2021.07.021. Epub 2021 Oct 12. PMID: 34649707

  • Fig. 3 - please add more photos, so that everyone could easily see the construction of the splint

we added the photo

  • In the discussion: add the aspect of "real bruxers" - the use of splint that touches only incisors could be a very unpleasent for them

Answer in Discussion in Occlusal Splint Production.

„Unfortunately, it is difficult for “real bruxers” to tolerate the device well while asleep. In general, splints that only touch the incisors are not indicated for TMD diseases as they act like dental braces and deform the occlusal plane.“

  • - refer to artificial aging of materials that could be used to manufacture splints, eg Dental LT Clear,.

We added information about artificial aging of materials from the relevant literature you recommended to the manuscript.

Paradowska-Stolarz, A.; Wezgowiec, J.; Malysa, A.; Wieckiewicz, M. Effects of Polishing and Artificial Aging on Mechanical Properties of Dental LT Clear® Resin. J. Funct. Biomater. 202314, 295. https://doi.org/10.3390/jfb14060295

  • Could this influence the long-term quality of the splint and its precission?

Answer in Discussion in Matarials.

„Dental LT Clear by Formlabs resin is a relatively new material and its biomechanical resistance properties are still the subject of research. The research has demonstrated that sterilization can yield benefits by reducing the elution of monomers while improving the microhardness of the resin. Specifically, au-toclaving at a temperature of 132°C for 4 minutes has been identified as particularly influential. [26] This is excellent news precisely because of the repeated use of anterior deprogrammers and during the prefabrication of occlusal splints.“

  • In the discussion, please add an important limitation, which is a limited possibility to use articulators when manufacturing any occlusal splint and how important that is - refer to commonly used facebows

Answer in Discussion in Prescription.

„A recent study compared the mechanical and computerized registration methods used by the two selected kinematic face bows. It found mechanical facebow handling has a higher risk of hand-measuring errors in tracing. The significant difference in the measurements of the condylar path inclination is most likely a result of the differences in the registration techniques assumptions. The authors recommend using an articula-tor compatible with a facebow whose measurement has been done. The authors as-sume that virtual tools for mandibular tracing with extremely high diagnostic poten-tial will soon be introduced to daily practice.“

  • I would add in the title, that the article is a literature review and presentation of the cases, because that is indeed what it is

Thank you very much for the valuable advice. We have changed the title.

„Literature review of an anterior deprogrammer to determine the centric relation and presentation of the cases“

Round 2

Reviewer 1 Report

Comments and Suggestions for Authors

Thanks for the response, I have no further comments

Author Response

Dear Reviewer,

we would like to thank you for your time, and we are happy to see all the changes we made based on your review and advice were satisfactory. Thank you very much!

Reviewer 3 Report

Comments and Suggestions for Authors

Dear authors,

The paper needs improvements.

In the abstract, still there are abbreviation without full name like CAD and 3D

In the discussion section bold sentences should be removed and you should make link between paragraphs. Where is the limitation?

The conclusion is still too long and it is not reflective to the work it should be rewritten in a short and more conclusive way.

Please add more references you change the title to literature so the references are not enough.

Comments on the Quality of English Language

Minor

Author Response

Dear Reviewer,

First and foremost, we would like to express our sincere appreciation for your time, valuable advice, and insightful feedback, which have significantly enhanced the quality of our manuscript. Thank you very much!

In the abstract, we included the full name along with the abbreviations.

We made several changes to the discussion section including removing bold sentences and adding links between paragraphs. Additionally, we included limitations:

- limitations in the recent studies (Introduction):

„Reviews often point out limitations in studies, which gives rise to the need for further research [10, 22, 23, 24, 25, 26, 27, 28]. These limitations may include small sample sizes, a lack of randomization, inadequate blinding methodologies, poor concealment of allocation, improper handling of withdrawals and losses, selective or incomplete reporting, short test periods, the absence of a control group, inadequate baseline data, and shortcomings in outcome reporting [10, 22, 24, 28]. The application of splints should therefore be considered at the individual patient level [29]. This may be due to the multifactorial nature of the disease. A thorough understanding of occlusion [30], phonetics, esthetics and laboratory steps is also necessary [25, 31].“

- limitations in related to production (discussion):

„Traditional techniques have limitations [47] such as dimensional inaccuracies caused by thermoplastics’ conventional thermoforming, which leads to discomfort and treatment failure [68, 47, 73]. Polymethyl methacrylate that is cured by heat is a commonly used material in the production of occlusal splints, which are useful in treating functional disorders of the stomatognathic system. It possesses many beneficial properties, including adequate hardness, low shrinkage level, chemical stability, resistance to abrasion [40], and easy processing and handling. Its cost-effectiveness is also improved as it is cheaper than light-cured resin. However, it’s not a perfect material as some individuals may experience hypersensitivity [76, 77] to the monomer present in acrylate materials. This can lead to skin allergies and respiratory allergies in both dental technicians and patients. Additionally, acrylate materials have a strong, pungent odor that disappears after polymerization [77, 78].“

Conclusion: We shortened the conclusion and rewrote it more convincingly.

Referencies: After conducting further research, we have included additional references to support our findings.

Sincerely, Authors

Reviewer 4 Report

Comments and Suggestions for Authors

Dear Authors, 

thank you for corrections.

Please, note that you have "Chyba! Nenašiel sa žiaden zdroj odkazov., Chyba! Ne- 62 našiel sa žiaden zdroj odkazov.," in several spots. Besides, use the shorts for references - the journals names (eg. 3 - Dent Med Probl)

Please, correct those errors. Besides, the paper is well enough to be accepted. Thank you

Author Response

Dear Reviewer,

we would like to thank you for your time, and we are happy to see all changes we made based on your review and advice were satisfactory. We corrected all mistakes. Again, thank you very much!